# Impact of Chest Trauma and Overweight on Mortality and Outcome in Severely Injured Patients

**DOI:** 10.3390/jcm9092752

**Published:** 2020-08-26

**Authors:** Thurid Eckhardt, Klemens Horst, Philipp Störmann, Felix Bläsius, Martijn Hofman, Christian Herren, Philipp Kobbe, Frank Hildebrand, Hagen Andruszkow

**Affiliations:** 1Department for Trauma and Reconstructive Surgery, University Hospital Aachen, Pauwelsstr. 30, 52074 Aachen, Germany; thurid.eckhardt@rwth-aachen.de (T.E.); khorst@ukaachen.de (K.H.); fblaesius@ukaachen.de (F.B.); mhofman@ukaachen.de (M.H.); cherren@ukaachen.de (C.H.); pkobbe@ukaachen.de (P.K.); fhildebrand@ukaachen.de (F.H.); 2Department for Trauma and Reconstructive Surgery, University Hospital Frankfurt am Main, Theodor-Stern-Kai 7, 60590 Frankfurt, Germany; philipp.stoermann@kgu.de

**Keywords:** chest trauma, weight disorders, overweight, obesity, outcome

## Abstract

The morbidity and mortality of severely injured patients are commonly affected by multiple factors. Especially, severe chest trauma has been shown to be a significant factor in considering outcome. Contemporaneously, weight-associated endocrinological, haematological, and metabolic deviations from the norm seem to have an impact on the posttraumatic course. Therefore, the aim of this study was to determine the influence of body weight on severely injured patients by emphasizing chest trauma. A total of 338 severely injured patients were included. Multivariate regression analyses were performed on patients with severe chest trauma (AIS ≥ 3) and patients with minor chest trauma (AIS < 3). The influence of body weight on in-hospital mortality was evaluated. Of all the patients, 70.4% were male, the median age was 52 years (IQR 36–68), the overall Injury Severity Score (ISS) was 24 points (IQR 17–29), and a median BMI of 25.1 points (IQR 23–28) was determined. In general, chest trauma was associated with prolonged ventilation, prolonged ICU treatment, and increased mortality. For overweight patients with severe chest trauma, an independent survival benefit was found (OR 0.158; *p* = 0.037). Overweight seems to have an impact on the mortality of severely injured patients with combined chest trauma. Potentially, a nutritive advantage or still-unknown immunological aspects in these patients affecting the intensive treatment course could be argued.

## 1. Introduction

Trauma is known to be a leading public health concern, with approximately 16,000 deaths worldwide due to various injuries each day [1]. In Western civilizations, severe trauma is the main cause of death among 15- to 44-year-olds [2]. Consequently, trauma contributes to the highest incidence of years of potential life lost, generating a large economic impact [1]. 

Severe chest trauma has been identified as one of the most frequent injury patterns, in addition to being a crucial prognostic factor and a leading cause of death after trauma [1]. Recent studies revealed extended periods of mechanical ventilation and length of ICU stay for severely injured patients with thoracic injuries [3]. Furthermore, severe chest trauma has been shown to be associated with the development of diverse complications (e.g., multiple organ dysfunction syndrome (MODS) and an increase of mortality by approximately 25% considering all trauma-related deaths) [3,4]. 

Meanwhile, obesity and overweight are constantly rising in modern society, leading to a growing public concern worldwide. Two-thirds of the adult population in the United States are known to be overweight or obese. In Europe, the incidence of obesity is increasing simultaneously, currently with 20% of European citizens considered to be obese [5]. Excess weight disorders represent risk factors for many diseases, such as hypertension, arthritis, heart disease, diabetes mellitus, and cancer. Even though the association between these chronic conditions and obesity has been investigated, varying results have been reported concerning the consequences of weight disorders on complications and outcome following trauma [6,7]. While some studies reveal increased mortality in underweight and obese patients, contemporary analyses indicate no impact of body weight [8,9]. Although these studies analysed the influence of weight disorders on outcome in trauma patients in general, little is known about the effects of body weight, particularly focusing on chest trauma as one of the most common injury patterns after trauma. 

Therefore, the present study intended to evaluate the impact of different weight entities on the posttraumatic course of traumatized patients with a focus on chest trauma.

## 2. Experimental Section

The present study follows the guidelines of the revised World Medical Association Declaration of Helsinki in 1975 and its latest amendment in 2013 (64th general meeting). Ethical approval was obtained from the Ethics Committee of the Medical Faculty of the RWTH Aachen University (EK 346/15). 

### 2.1. Study Design and Inclusion Criteria

A retrospective analysis of traumatized patients admitted to the RWTH Aachen University Hospital (level I trauma centre) between 2010 and 2015 was performed. Hospital charts were studied to acquire standardised documentation of the clinical course from first contact with the responsible physicians until discharge or death in the hospital. The collected data included information on demographics, pattern of injury, comorbidities, preclinical and clinical management, procedures, laboratory findings, and outcome. To ensure the highest possible data quality, all parameters were immediately reviewed for plausibility. The inclusion criteria for the present study were an Injury Severity Score (ISS) ≥ 9 and patient age ≥ 18 years. Minor injury patients (ISS < 9) aged < 18 and patients who deceased on-scene were excluded. 

### 2.2. Injury Distribution and Injury Severity

To evaluate the thoracic injury distribution, the 2005 revised and 2008 updated versions of the Abbreviated Injury Scale (AIS) were used [10]. The AIS is an anatomy-based severity scoring system that classifies injuries by body region. Each injury is assigned an AIS score on a six-point scale according to importance, from 1 (minor) to 6 (unsurvivable) [11]. In the present study, relevant severe chest trauma was defined by an AIS score ≥ 3, while minor chest trauma was defined by an AIS score < 3. Consequently, patients without chest trauma (AIS = 0) were included in this group. 

The AIS score is the basis for the calculation of the Injury Severity Score (ISS), determining the overall injury distribution. To calculate the ISS, identification of the three most seriously injured body regions is necessary. The highest AIS score for each of these regions is squared and summed up to a total score [10]. 

Furthermore, multiple prognostic scores were compared with the observed outcome to allow the examination of different injury patterns and severity between the respective subgroups. The investigated scores contained the anatomic injury-based New Injury Severity Score (NISS) [12].

### 2.3. In-Hospital Complications and Mortality

The severity of multiple organ dysfunction was measured using the Multiple Organ Dysfunction Score (Marshall MODS) [13]. This score mirrors organ dysfunction by using simple physiologic measures of dysfunction in six organ systems. It correlates with the highest risk of ICU and hospital mortality [13]. Additionally, the total number of deaths during the entire clinical stay was determined to note the in-hospital mortality.

### 2.4. Body Mass Index (BMI)

Weight disorders were assessed using the body mass index (BMI), an index of weight for height, which is defined as weight in kilograms divided by the square of height in metres (kg/m^2^) [14]. The population was subsequently classified into four groups based on the BMI using the classification of the World Health Organization as follows: BMI < 20 kg/m^2^ (underweight), BMI 20–25 kg/m^2^ (normal weight), BMI 25–30 kg/m^2^ (overweight), and BMI > 30 kg/m^2^ (obesity). Body weight and height were measured at hospital admission.

### 2.5. Statistical Analysis

Data were analysed using the Statistical Package for the Social Sciences (SPSS 25, IBM Inc., Somers, NY, USA). Continuous values are presented as median with 25 and 75 interquartile ranges (25/75-IQR), while developments are presented with counts and percentages. Differences between the groups were evaluated with nonparametric Mann–Whitney U tests for continuous data, while Pearson’s chi-square test was used for categorical values. To verify the impact of chest trauma and weight disorders on outcome, multivariate logistic regression analyses were performed. Odds ratios (ORs) with 95% confidence interval (95%-CI) were noted. A two-sided *p*-value < 0.05 was considered to be significant.

## 3. Results

A total of 338 patients admitted to the RWTH University Hospital between 2010 and 2015 fulfilled the inclusion criteria. Table 1 provides a demographic overview according to the presence of severe chest trauma, including data on age, gender, BMI, injury distribution, and mortality. In general, the median age was 52 years (IQR 36–68), and 238 patients were male (70.4%). In total, 92.3% suffered from blunt and 7.7% from penetrating injuries, and the most common cause was traffic accident (66.4%). 

### 3.1. Chest Trauma

A total of 137 patients with severe chest trauma (group I) and 201 patients with minor chest trauma (group II) were analysed. 

### 3.2. Impact of BMI and Chest Trauma

Multivariate logistic regression analysis revealed a survival benefit for patients with severe chest trauma in association with overweight compared with patients with only minor chest trauma and obesity or normal weight (Table 2: OR 0.158; *p* = 0.037). Furthermore, the severity of traumatic brain injury was shown to have an impact on mortality (OR 1.822, *p* = 0.001). Age, gender, and the overall injury severity had no influence on mortality, according to this statistical model.

In contrast, the aforementioned survival benefit of overweight could not be noted for patients with only minor chest trauma (Table 3). For these patients, age and injury severity were determined as influencing factors for mortality.

## 4. Discussion 

The present study was designed to evaluate the influence of different weight entities on outcome in severely injured patients, with emphasis on chest trauma. To the best of our knowledge, the presented results were the first to focus on this specific injury pattern in multiple trauma patients. 

Our main results can be summarized as follows:Severe chest trauma was associated with prolonged treatment and in-hospital complications.Subgroup analysis revealed overweight to be an independent survival factor in patients with severe chest trauma. In contrast, this aspect was not proven for severely injured patients with only minor chest trauma.

The treatment of trauma patients represents a clinical challenge, particularly in the case of associated severe chest trauma. Epidemiologic data indicate a high coincidence of severe chest trauma in severely injured patients, with consistent rates during the last several decades [15,16]. 

Analysing patients with only minor chest trauma, we found a positive correlation between mortality and Glasgow Coma Scale (GCS), AIS head, ISS, and age. Overall mortality in this group was slightly elevated compared with that in a corresponding study by Bayer et al. [17]. The reason could be the exclusion of patients with severe head injury in their analysis. Regarding the mean ISS for multiple trauma patients with an AIS chest < 3, no significant difference could be detected. Additionally, the total length of stay and the limited need for mechanical ventilation in patients without severe chest trauma compared with minor chest trauma patients coincided.

A recently performed prospective study by Grubmüller et al. investigated the impact of severe chest trauma on severely injured patients, showing a significantly increased rate of in-hospital complications like organ failure and respiratory failure [18]. Consistent with that analysis, the presented study found increased incidence of MODS (*p* < 0.001; r = 0.849). Similar results were also found in several other studies examining the influence of chest trauma on multiple organ dysfunction [16,19]. Respiratory insufficiency and hemodynamic failure represent the major causes of deaths in patients with severe chest trauma, resulting in mortality rates of up to 25% [20,21]. This underlines the results of other studies showing that chest trauma is one of the most important concomitant injuries in severely injured patients having an impact on mortality [16,19,22]. Accordingly, we also found a significantly higher mortality rate for patients with an AIS chest ≥ 3 compared with patients with an AIS chest < 3. However, not all studies found an influence of severe chest trauma on mortality [18]. Potential reasons for these divergent results might be the exclusion of all patients with penetrating chest trauma and a lower overall mortality rate in the study of Grubmüller et al. [18]. The aforementioned impact of severe chest trauma on the posttraumatic clinical course is also mirrored by a prolongation of both duration of ventilation and ICU treatment in our study. In accordance with our results, Bayer et al. also revealed a correlation between an increased AIS chest score and a prolonged intubation period (*p* = 0.005) [17]. Moreover, Lin et al. found a significant association between severe chest trauma and prolonged ICU treatment (*p*  <  0.003) [23]. It might be summarized that the results of the presented study on the impact of chest trauma on mortality and the clinical course are in line with those of the vast majority of studies in the current literature. 

Overweight and obesity are steadily increasing in modern society. In accordance with international data, only 45.6% of our study population presented with a normal weight [24]. In general, a BMI > 30 has been described as increasing the risk for the development of multiple organ failure with an associated prolongation of the length of ICU and overall in-hospital treatment [25]. Relevant medical comorbidities related to weight disorders have been supposed to be the most likely risk factors for postsurgical and posttraumatic complications [26]. A prospective study by Goulenok et al. analysed severely injured patients, focusing on two body weight groups, BMI < 25 and >25. The authors revealed an increased mortality during ICU treatment for the second group [27]. In this respect, the present study generated four study groups according to the established BMI definition in order to investigate the impact of different weight entities on mortality [28]. We feel it is safe to argue that the current WHO classification offers a more precise predictive validity compared with the aforementioned BMI graduation by Goulenok [25]. In this context, the presented subgroup analysis revealed overweight (BMI 25–30) to be an independent survival factor in severely injured patients with concomitant chest trauma. This main result is supported by previous observations by Mica et al., who found a protective effect of a BMI between 25 and 30 points on the Systemic Inflammatory Response Syndrome (SIRS) and sepsis after severe trauma in general. Furthermore, they found the mortality rate to be decreased in the overweight BMI group compared with that in normal-weight patients (7.2% vs. 8.8%; *p* < 0.001). Significant differences in the clinical course, like duration of ICU treatment, duration of ventilation, and overall hospital stay, were not found. In contrast to the presented study, Mica et al. did not focus on a specific injury distribution like chest trauma [29,30]. The present study could therefore be argued to focus on potential outcome variables after trauma being more valid, with greater emphasis on different injury patterns and weight entities. 

However, currently, only Fatica et al. investigated a potential association between chest trauma and body mass index in severely injured patients [22]. In their retrospective study with 233 thoracic trauma patients, obesity (defined as a BMI > 25) increased mortality independently of overall injury severity in chest trauma patients. Additionally, hospital admission rate, length of hospital stay, and injury severity were significantly increased for patients with a BMI > 25 [22]. However, as the present study defined overweight and obesity more precisely compared with Fatica et al., we were able to specify these results, indicating that obesity but not overweight seems to be associated with adverse outcome, whereas overweight was identified as an independent survival factor after severe trauma with severe chest trauma. In this context, a nutritive advantage for overweight patients during the prolonged intensive care course might represent a feasible explanation [31]. 

Additionally, a shielding impact of fatty tissue on inflammatory reactions following trauma, supressing an excessive immune response, should be considered. The reason for this could be the protective effect of oestrogens produced by aromatase activity in adipocytes as well as the immunomodulatory effect of adiponectin, which can lead to endotoxin tolerance and lower susceptibility to generalized inflammation [32,33]. Whilst patients with a BMI > 30 often suffer from chronic medical comorbidities and present with higher mortality rates after severe trauma, overweight patients with a BMI ranging from 25 to 30 points rarely present with severe medical conditions and can therefore benefit from the protective effect of the additional fatty tissue.

In patients with only minor chest trauma, no advantage for overweight patients was noted. The reason for this could be the different pattern of injury and different injury characteristics for the respective weight entities [34,35]. Additionally, the lung represents a primary target organ due to the inflammatory response after multiple trauma and could therefore be more predisposed to secondary damage in patients with chest trauma [15]. Studying the obesity-related inflammatory profile could provide additional insight. 

Interpretation of the presented results should consider the limitations of this study. First, this study is limited by its retrospective design. Furthermore, BMI was the only parameter used for body weight since it represents the most widely accepted parameter in current literature analysing trauma populations [7,36]. Our two groups (AIS ≥ 3 vs. AIS < 3) differed considerably on the extent of injury severity, measured by the ISS (ISS 27 ± 11 Gr. I vs. 23 ± 8 Gr. II, *p* < 0.001) and the incidence of head trauma (AIS 1.7 ± 2 Gr. I vs. 3.4 ± 2 Gr. II, *p* < 0.001). In order to consider the divergent groups statistically, a multivariate regression analysis was performed. 

Notwithstanding these limitations, this study offers new aspects regarding morbidity and mortality in severely injured patients, emphasizing different weight entities with concomitant chest trauma. In this respect, a BMI ranging from 25 to 30 seems to alter the clinical course, and its protective effects must be investigated in future studies.

## 5. Conclusions

Based on the presented analysis, the presence of severe chest trauma in severely injured patients has a considerable impact on the clinical course. Furthermore, divergent weight entities seem to affect this impact on outcome. The treatment of severely injured patients should therefore reflect the presence of overweight or obesity, assessing this influencing factor correctly during the clinical course. 

## Figures and Tables

**Table 1 jcm-09-02752-t001:** Demographic overview according to the presence of severe chest trauma.

	Overall	Group I	Group II	*p*-Value
Number of patients (*n*)	338	137	201	-
Age (median 25/75-IQR)	52 (36–68)	49.0 (38–62)	52 (34–72)	0.175
Male (%)	70	75	67	0.113
ISS (median 25/75-IQR)	24 (17–29)	27.0 (18–34)	24 (17–25)	<0.001
AIS head (median 25/75-IQR)	3.5 (0–4)	1.0 (0–3)	4.0 (3.0–5.0)	<0.001
BMI (median 25/75-IQR)	25.1 (23–28)	25.7 (24–28)	24.8 (23–28)	0.042
Underweight (*n*)	30	13	17	
Normal weight (*n*)	146	54	92	
Overweight (*n*)	119	53	66	
Obesity (*n*)	43	20	23	
MODS (%)	26.3	30.3	20.1	<0.001
Mortality (%)	24.3	32.3	12.4	<0.001
Duration of ventilation (h) (median 25/75-IQR)	48 (2–258)	70 (10–222)	25 (0–287)	0.022
Duration of ICU treatment (days) (median 25/75-IQR)	11 (1–23)	17 (9–29)	6 (2–17)	<0.001
Length of stay (days) (median 25/75-IQR)	16 (7–27)	17 (10–29)	14 (5–25)	0.008

IQR: Interquartile range; BMI: Body mass index; MODS: Multiple Organ Dysfunction Score (Marshall MODS).

**Table 2 jcm-09-02752-t002:** Multivariate regression analysis of patients with Abbreviated Injury Scale (AIS) chest ≥ 3 referring to mortality.

	Regression Coefficient	Odds Ratio (OR)	95% Confidence Interval (95%-CI)	*p*-Value
Age (years)	0.027	1.028	0.988–1.069	0.172
Gender (male)	−0.218	0.804	0.232–6.671	0.799
Overweight	−1.847	**0.158**	**0.028–0.892**	**0.037**
Obesity	0.030	1.094	0.203–5.885	0.917
ISS (per point)	0.057	1.059	0.989–1.133	0.102
AIS head	0.600	**1.822**	**1.102–3.013**	**0.019**
Constant	−6.148	−	−	<0.001

Normal weight was set as a categorical reference group for regression analysis between the BMI groups. The significant variables were in bold.

**Table 3 jcm-09-02752-t003:** Multivariate regression analysis of patients with AIS chest < 3 referring to mortality.

	Regression Coefficient	Odds Ratio (OR)	95% Confidence Interval (95%-CI)	*p*-Value
Age (years)	**0.028**	**1.028**	**1.009–0.048**	**0.004**
Gender (male)	0.387	1.472	0.684–3.166	0.323
Overweight	−0.311	0.733	0.334–1.607	0.438
Obesity	−0.282	0.755	0.234–2.434	0.637
ISS (per point)	**0.068**	**1.071**	**1.023–1.120**	**0.003**
AIS head	0.341	1.407	1.096–1.806	0.007
Constant	−4.928	-	-	0.007

Normal weight was set as a categorical reference group for regression analysis between the BMI groups. The significant variables were in bold.

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
