# Peer review of "Impact of Chest Trauma and Overweight on Mortality and Outcome in Severely Injured Patients"

_jcm, 2020, doi:10.3390/jcm9092752_

Round 1

Reviewer 1 Report

Overall, a reasonable retrospective cohort study investigating the association between obesity and severe chest trauma. The investigators found obesity to be associated with a significant survival benefit by multivariable logistic regression.

I see inclusion criteria listed in Section 2.1. However, I do not see exclusion criteria. If there were no exclusion criteria – state this. Be extremely clear about how “Group II” was defined. I see this group had an AIS <3 but does this mean zero chest trauma or minor chest trauma? Do the researchers consider this group more of a “control”? Please clarify

Table 1: Consider labeling “Group I” as the “AIS ≥3 or severe chest trauma” “Group II” as “AIS <3 or minor (or zero) chest trauma” so the reader can easily reference which is which without searching through the text

The finding that obesity is correlated with improved clinical outcomes is interesting; particularly because others have identified similar correlations.

Consider a more thorough analysis about why obesity might be predictive in the Discussion section. Perhaps discuss beyond “a nutritive advantage for overweight patients…”

Author Response

Impact of chest trauma and overweight on mortality and outcome in severely injured patients

Thurid Eckhardt1, Klemens Horst 1, Philipp Störmann2, Felix Bläsius1, Martijn Hofman1, Christian Herren1, Philipp Kobbe1, Frank Hildebrand1, Hagen Andruszkow1

1Department for Trauma and Reconstructive Surgery, University Hospital Aachen, Pauwelsstr. 30, 52074 Aachen, Germany; [email protected]; [email protected]; [email protected]; [email protected]; [email protected]; [email protected]; [email protected]; [email protected]

2 Department for Trauma and Reconstructive Surgery, University Hospital Frankfurt am Main, Theodor-Stern-Kai 7, 60590 Frankfurt, Germany; [email protected]

Correspondence: [email protected]; Tel: +49-(0)241-80-36076

Authors’ notes to reviewer 1:

Overall, a reasonable retrospective cohort study investigating the association between obesity and severe chest trauma. The investigators found obesity to be associated with a significant survival benefit by multivariable logistic regression.

I see inclusion criteria listed in Section 2.1. However, I do not see exclusion criteria. If there were no exclusion criteria – state this. Be extremely clear about how “Group II” was defined. I see this group had an AIS <3 but does this mean zero chest trauma or minor chest trauma? Do the researchers consider this group more of a “control”? Please clarify

Answer: Group II was defined as minor chest trauma which also included patients without chest trauma. This group was meant as a statistical control group compared to patients with considerable chest trauma. We decided to use this definition as it is well established focusing results of nationwide trauma registries (e.g. PLoS One. 2017; 12(10): e0186712.; PLoS One. 2016; 11(1): e0146897.)

As exclusion criteria we defined an ISS <9 points, age <18 as well as patients who deceased on-scene. As answered above, patients without chest trauma were not excluded and implemented to Group II which was meant as minor chest trauma group.

The following aspect regarding exclusion criteria was added. Lines 69 – 70: Minor injured patients (ISS<9), age <18 as well as patients who deceased on-scene were excluded.

Furthermore, we added the following aspect to the method section: lines 75 – 777: In the present study, relevant severe chest trauma was defined by an AIS Score ≥3 while minor chest trauma was defined as AIS <3. In consequence, patients without chest trauma (AIS = 0) were included to this group.

Table 1: Consider labelling “Group I” as the “AIS ≥3 or severe chest trauma” “Group II” as “AIS <3 or minor (or zero) chest trauma” so the reader can easily reference which is which without searching through the text

Answer: We changed the labelling and named “Group I” as “severe chest trauma” and “Group II” as “minor chest trauma” throughout the manuscript.

The finding that obesity is correlated with improved clinical outcomes is interesting; particularly because others have identified similar correlations.

Consider a more thorough analysis about why obesity might be predictive in the Discussion section. Perhaps discuss beyond “a nutritive advantage for overweight patients…”

Answer: We added the following paragraph to the discussion section in order to field the aspect “beyond nutritive advantages”:

Lines 207-214: Also a shielding impact of fatty tissue on inflammatory reactions following trauma, supressing an excessive immune response, should be considered. Reason for this could be the protective effect of oestrogens produced by aromatase activity in adipocytes as well as the immunomodulatory effect of adiponectin which can lead to endotoxin tolerance and lower susceptibility to generalized inflammation [32,33]. Whilst patients with a BMI >30 often suffer from chronic medical comorbidities and present with higher mortality rates after severe trauma, overweight patients with a BMI ranging from 25-30 points, rarely present with severe medical conditions and can therefore benefit from the protective effect of the additional fatty tissue.

Reviewer 2 Report

Summary: This manuscript is a retrospective analysis of 338 trauma patients admitted to a single level 1 trauma center in Germany between 2010 & 2015. Hospital records were abstracted for all patients with ISS>=9 and age >=18 years. Descriptive statistics as well as multivariable logistic regressions are presented to “determine the influence of body weight on severely injured patients emphasizing chest trauma.”

General comments:

  • Grammatical errors, typically in plurals, article usage or verb tense, are present in nearly every paragraph. Editing for English grammar and flow would improve this manuscript.
  • The aim of the analysis is a little unclear. The terms “chest trauma” and “severe chest trauma” are used interchangeably throughout the manuscript, confusing the reader. Please clearly define your population of interest and use 1 term exclusively. It is also unclear whether or not patients with no chest injury were included in the analysis because it is not listed as an exclusion criterion in the methods section. A consort diagram made be useful if these patients were excluded from the analysis.
  • The analysis plan and presentation are very underdeveloped. This manuscript needs to have a clear statement of the question defined, which would then guide the analysis. More involvement by a statistician would be warranted.

Specific comments:

  • Are patients without chest injuries (e.g. AIS chest=0 or missing) excluded from the analysis? They are not mentioned anywhere in the manuscript. If patients without a chest injury were excluded, this should be noted in the methods.
  • I am concerned that data for age, AIS head and ISS are skewed and would better be represented by medians with interquartile ranges rather than means and standard deviations. Are the distributions for these variables skewed and if so, please justify the presentation of means and the use of ANOVA rather than a non-parametric test?
  • Please present the appropriate measures of central tendency and variability for duration of ventilation, duration of ICU treatment, and length of stay in table 1.
  • The rationale for presenting Spearman correlation coefficients in table 2 is unclear. Severe chest trauma is dichotomized in table 1; what is the severe chest trauma variable being used to correlate with outcomes in table 2? Is the monotonic assumption for the Spearman’s correlation met for all variables? If vent hours, ICU days, and LOS were included in table 1 and compared between groups, table 2 could be deleted. I would prefer to see unadjusted and adjusted regressions rather than simple correlations to support the first conclusion drawn.
  • Please describe how covariates/confounders in the 2 multivariate logistic regressions were chosen. Were any variables considered exclusively in 1 model and not the other? Do the authors believe that predictors may be different between the 2 models and if not, why present 2 models? Another alternative would be to present 1 model, including a dichotomous variable for AIS chest >=3. Another would be to only analyze the patients with AIS chest >=3 if the question you want to answer is how does overweight impact mortality in the presence of head trauma.
  • Pseudo R2s, like the Nagelkerke R2, are often not presented in manuscripts for a clinical audience b/c they cannot be interpreted in the same way as an R2 from ordinary least square regression. I would recommend not presenting the Nagelkerke R2 in this manuscript, especially without discussion of it.
  • It is unclear how BMI is included in the logistic regression models. Are overweight and obesity included in the models as dichotomous variables only? The form of each variable should be clearly presented in a footnote, especially if included in a different form than described in the methods. Additionally, how many patients are overweight and obese? Because information regarding BMI in table 1 is presented as means, it is difficult to determine how many patients are included in each BMI category. The wide confidence intervals for obesity in both models indicate that relatively few patients are obese. Why is obesity included in either model- this goes back to my question regarding how variables were chosen for the model.
  • Was there any missingness in any of the variables?
  • The first main result listed in line 143 is not clearly supported by the data because the variable for ‘severe chest trauma” is not defined in table 2. This would be a true statement only if patients without chest trauma were included in the correlation. A more supported statement (assuming AIS chest was used as an ordinal variable) would be “severity of chest trauma was associated with prolonged treatment...”
  • The paragraph included in lines 157-176 is irrelevant to a discussion of overweight/obesity and chest trauma. Shorten or delete

Author Response

Impact of chest trauma and overweight on mortality and outcome in severely injured patients

Thurid Eckhardt1, Klemens Horst 1, Philipp Störmann2, Felix Bläsius1, Martijn Hofman1, Christian Herren1, Philipp Kobbe1, Frank Hildebrand1, Hagen Andruszkow1

1Department for Trauma and Reconstructive Surgery, University Hospital Aachen, Pauwelsstr. 30, 52074 Aachen, Germany; [email protected]; [email protected]; [email protected]; [email protected]; [email protected]; [email protected]; [email protected]; [email protected]

2 Department for Trauma and Reconstructive Surgery, University Hospital Frankfurt am Main, Theodor-Stern-Kai 7, 60590 Frankfurt, Germany; [email protected]

Correspondence: [email protected]; Tel: +49-(0)241-80-36076

Authors’ notes to reviewer 2:

General comments:

  • Grammatical errors, typically in plurals, article usage or verb tense, are present in nearly every paragraph. Editing for English grammar and flow would improve this manuscript.
  • The aim of the analysis is a little unclear. The terms “chest trauma” and “severe chest trauma” are used interchangeably throughout the manuscript, confusing the reader. Please clearly define your population of interest and use 1 term exclusively. It is also unclear whether or not patients with no chest injury were included in the analysis because it is not listed as an exclusion criterion in the methods section. A consort diagram made be useful if these patients were excluded from the analysis.
  • The analysis plan and presentation are very underdeveloped. This manuscript needs to have a clear statement of the question defined, which would then guide the analysis. More involvement by a statistician would be warranted.

Answer: Thank you very much for the important critical aspects.  To improve English grammar and flow a native speaker editing was performed.

We now clearly defined the two study groups. Group II was defined as minor chest trauma which also included patients without chest trauma. This group was meant as a statistical control group compared to patients with considerable chest trauma. These patients were now named as “severe chest trauma” throughout the manuscript. We decided to use this definition as it is well established focusing results of nationwide trauma registries (e.g. PLoS One. 2017; 12(10): e0186712.; PLoS One. 2016; 11(1): e0146897.)

As exclusion criteria we defined an ISS <9 points, age <18 as well as patients who deceased on-scene. As answered above, patients without chest trauma were not excluded and implemented to Group II which was meant as minor chest trauma group.

The following aspect regarding exclusion criteria was added. Lines 69 – 70: Minor injured patients (ISS<9), age <18 as well as patients who deceased on-scene were excluded.

Furthermore, we added the following aspect to the method section: lines 75 – 77: In the present study, relevant severe chest trauma was defined by an AIS Score ≥3 while minor chest trauma was defined as AIS <3. In consequence, patients without chest trauma (AIS = 0) were included to this group.

Specific comments:

  • Are patients without chest injuries (e.g. AIS chest=0 or missing) excluded from the analysis? They are not mentioned anywhere in the manuscript. If patients without a chest injury were excluded, this should be noted in the methods.

Answer: We clarified this in the method section: Furthermore, we added the following aspect to the method section: lines 75 – 77: In the present study, relevant severe chest trauma was defined by an AIS Score ≥3 while minor chest trauma was defined as AIS <3. In consequence, patients without chest trauma (AIS = 0) were included to this group.

  • I am concerned that data for age, AIS head and ISS are skewed and would better be represented by medians with interquartile ranges rather than means and standard deviations. Are the distributions for these variables skewed and if so, please justify the presentation of means and the use of ANOVA rather than a non-parametric test?

Answer: We performed non-parametric Mann Whitney U tests for continuous data now. Please find the changes highlighted in the tables.

  • Please present the appropriate measures of central tendency and variability for duration of ventilation, duration of ICU treatment, and length of stay in table 1.

Answer: Please find the new data added to table 1.

  • The rationale for presenting Spearman correlation coefficients in table 2 is unclear. Severe chest trauma is dichotomized in table 1; what is the severe chest trauma variable being used to correlate with outcomes in table 2? Is the monotonic assumption for the Spearman’s correlation met for all variables? If vent hours, ICU days, and LOS were included in table 1 and compared between groups, table 2 could be deleted. I would prefer to see unadjusted and adjusted regressions rather than simple correlations to support the first conclusion drawn.

Answer: As the reviewer recommended above, we implemented the requested new data to table 1 and deleted table 2. Our intention was to demonstrate the clinically relevant association between severe chest trauma and enhanced treatment durations. As now demonstrated in table 1, the impact of chest trauma can now be seen in table 1.

  • Please describe how covariates/confounders in the 2 multivariate logistic regressions were chosen. Were any variables considered exclusively in 1 model and not the other? Do the authors believe that predictors may be different between the 2 models and if not, why present 2 models? Another alternative would be to present 1 model, including a dichotomous variable for AIS chest >=3. Another would be to only analyze the patients with AIS chest >=3 if the question you want to answer is how does overweight impact mortality in the presence of head trauma.

Answer: We chose the covariates from a clinical view. Demographic influences as well as injury severity are commonly known as confounders regarding outcome in trauma patients. The same co-variables were used in both regression models. We decided to present two regression analyses to clarify the impact in both chest trauma groups. In our opinion, presenting only one model with severe chest trauma would be possible but impair the clinical view on both groups.

  • Pseudo R2s, like the Nagelkerke R2, are often not presented in manuscripts for a clinical audience b/c they cannot be interpreted in the same way as an R2 from ordinary least square regression. I would recommend not presenting the Nagelkerke R2 in this manuscript, especially without discussion of it.

Answer: We deleted the Nagelkerke R2 in this manuscript.

  • It is unclear how BMI is included in the logistic regression models. Are overweight and obesity included in the models as dichotomous variables only? The form of each variable should be clearly presented in a footnote, especially if included in a different form than described in the methods. Additionally, how many patients are overweight and obese? Because information regarding BMI in table 1 is presented as means, it is difficult to determine how many patients are included in each BMI category. The wide confidence intervals for obesity in both models indicate that relatively few patients are obese. Why is obesity included in either model- this goes back to my question regarding how variables were chosen for the model.

Answer: As described in the method section, we classified BMI into four groups (lines 94 – 96): The population was subsequently classified into four groups based on the BMI using the classification of the World Health Organization as follows: BMI <20 kg/m² (underweight), BMI 20-25 kg/m² (normal weight), BMI 25-30 kg/m² (overweight) and BMI >30 kg/m² (obesity). Normal weight was set as a categorical reference group for regression analysis between the BMI groups of overweight and obesity.

Furthermore, we added the number of patients according to BMI and chest trauma to table 1.

  • The first main result listed in line 143 is not clearly supported by the data because the variable for ‘severe chest trauma” is not defined in table 2. This would be a true statement only if patients without chest trauma were included in the correlation. A more supported statement (assuming AIS chest was used as an ordinal variable) would be “severity of chest trauma was associated with prolonged treatment...”

Answer: The exact definition of minor chest trauma is now implemented to the method section. Patients without chest trauma were included to the minor chest trauma group.

  • The paragraph included in lines 157-176 is irrelevant to a discussion of overweight/obesity and chest trauma. Shorten or delete

Answer: We intended to demonstrate that the presented data is in line with already established results from divergent international studies. We shortened this aspect.

Round 2

Reviewer 2 Report

Thank you for addressing my previous comments. I have 2 more items to check:

1) line 100: You state that means and Std Dev will be presented but medians and IQRs were presented in table 1. Also means are still presented in the abstract. Please convert to medians so the text and tables will be congruent with the abstract

2) table 1: the p-value for vent hours looks strange (00.22). Please double check this value

Author Response

Authors’ notes to reviewer 2:

2nd round:

Thank you for addressing my previous comments. I have 2 more items to check:

1) line 100: You state that means and Std Dev will be presented but medians and IQRs were presented in table 1. Also means are still presented in the abstract. Please convert to medians so the text and tables will be congruent with the abstract

2) table 1: the p-value for vent hours looks strange (00.22). Please double check this value.

Answer: Thank you very much for the second review. We excuse for the remaining aspects. All new changes are now highlighted in green.

1) line 100: “Continuous values are presented as median with 25- and 75- interquartile ranges (25/75-IQR) ….”

We changed this part of the statistic section stating now that continuous values are now presented as requested. Furthermore, the abstract was changed demonstrating median and interquartile ranges as required: lines 23 - 24: “ Results: 70.4% of all patients were male gender, median age was 52 years (IQR 36 – 68), overall ISS was 24 points (IQR 17 – 29), and a median BMI of 25.1 points (IQR 23 – 28) was determined.” In addition, lines 111-112 were changed in accordance. Text and tables are now congruent.

2) table 1: This was a mistake. We changed to the correct p-value of 0.022.